# Induction Chemotherapy Followed by Pleurectomy Decortication and Hyperthermic Intraoperative Chemotherapy (HITHOC) for Early-Stage Epitheliod Malignant Pleural Mesothelioma—A Prospective Report

**DOI:** 10.3390/jcm10235542

**Published:** 2021-11-26

**Authors:** Stefano Bongiolatti, Francesca Mazzoni, Ottavia Salimbene, Enrico Caliman, Carlo Ammatuna, Camilla E. Comin, Lorenzo Antonuzzo, Luca Voltolini

**Affiliations:** 1Thoracic Surgery Unit, Careggi University Hospital, 50134 Florence, Italy; salimbeneottavia@gmail.com (O.S.); luca.voltolini@unifi.it (L.V.); 2Medical Oncology Unit, Careggi University Hospital, 50134 Florence, Italy; francescamazzoni@hotmail.com (F.M.); enrico.caliman@gmail.com (E.C.); lorenzo.antonuzzo@gmail.com (L.A.); 3Department of Experimental and Clinical Medicine, Section of Surgery, Histopathology and Molecular Pathology, University of Florence, 50134 Florence, Italy; drcarloammatuna@gmail.com (C.A.); camilla.comin@unifi.it (C.E.C.); 4Department of Experimental and Clinical Medicine, University of Florence, 50134 Florence, Italy

**Keywords:** epithelioid malignant pleural mesothelioma, pleurectomy and decortication, hyperthermic intraoperative thoracic chemotherapy, overall survival, disease-free survival, progression-free survival

## Abstract

Malignant pleural mesothelioma (MPM) is an aggressive disease with poor prognosis and the current treatment for early-stage MPM is based on a multimodality therapy regimen involving platinum-based chemotherapy preceding or following surgery. To enhance the cytoreductive role of surgery, some peri- or intra-operative intracavitary treatments have been developed, such as hyperthermic chemotherapy, but long-term results are weak. The aim of this study was to report the post-operative results and mid-term outcomes of our multimodal intention-to-treat pathway, including induction chemotherapy, followed by surgery and Hyperthermic Intraoperative THOracic Chemotherapy (HITHOC) in the treatment of early-stage epithelioid MPM. Since 2017, stage I or II epithelioid MPM patients have been inserted in a surgery-based multimodal approach comprising platinum-based induction chemotherapy, followed by pleurectomy and decortication (P/D) and HITHOC with cisplatin. The Kaplan–Meier method was used to estimate overall survival (OS), disease-free survival (DFS) and progression-free survival (PFS). During the study period, *n* = 65 patients affected by MPM were evaluated by our institutional Multidisciplinary Tumour Board; *n* = 12 patients with stage I-II who had no progression after induction chemotherapy underwent P/D and HITHOC. Post-operative mortality was 0, and complications developed in *n* = 7 (58.3%) patients. The median estimated OS was 31 months with a 1-year and 3-year OS of 100% and 55%, respectively. The median PFS was 26 months with 92% of a 1-year PFS, whereas DFS was 19 months with a 1-year DFS rate of 83%. The multimodal treatment of early-stage epithelioid MPM, including induction chemotherapy followed by P/D and HITHOC, was well tolerated and feasible with promising mid-term oncological results.

## 1. Introduction

Malignant pleural mesothelioma (MPM) is an aggressive, fatal disease with a very poor prognosis [1]. The optimal therapeutic approach for early-stage MPM is not universally standardised yet, mainly due to the peculiarity of the pathology, to the imprecise clinical staging methods and to the unfeasibility to obtain a surgical complete resection. In particular, the role of surgery is still under great debate. Since surgical cytoreduction is not expected to yield an R0 resection, surgery has to be part of a multimodality treatment, including chemotherapy and/or radiotherapy. For early stage MPM, current treatment options always integrate multimodality concepts, include chemotherapy [2] before or after surgery [3], surgery with or without intraoperative local treatments, radiation therapy and, more recently, immunotherapy [4,5,6]. Although surgery still remains the cornerstone in the multimodal management of early-stage MPM, its precise role and type are highly controversial. There has been a general shift in surgery for MPM from extra pleural pneumonectomy (EPP) to pleurectomy and decortication (P/D), because more evidence is now available in favour of lung-sparing techniques [7,8,9,10]. Due to the diffusely infiltrative nature of MPM, the aim and possibility, at best, of surgery are to achieve a complete macroscopic resection [11], resulting, in most cases, in local relapse. Consequently, several other kinds of local treatments have been added to surgery in order to improve the local control and then long-term survival. Recently, few studies have shown favourable post-operative outcomes from the association between cytoreductive surgery and intra-operative intracavitary treatments, such as hyperthermic chemotherapy, hyperthermic povidone iodine and photodynamic therapy [12,13,14,15,16]. However, the long-term results of these integrated approaches suffered from the typical biases of single-centre retrospective analyses, with the inclusion of patients treated on and off protocols during the long study period.

In 2017, we started a new treatment protocol for clinical stage I and II epithelioid MPM patients, which includes induction chemotherapy followed by surgery and Hyperthermic Intraoperative THOracic Chemotherapy (HITHOC). 

The aim of this study is to report the post-operative results and mid-term outcomes of our recent experience with the multimodality approach from 2017 to 2020.

## 2. Materials and Methods

From January 2017 to December 2020, all patients affected by clinical stage I or II epithelioid MPM were inserted in a surgery-based multimodal approach consisting of platinum-based iCT [2] followed by P/D and HITHOC with cisplatin [13]. Data from these patients treated at Careggi University Hospital have been prospectively collected into an institutional database and retrospectively reviewed and analysed. Clinical stage was assessed by whole-body Computed Tomography (CT) scan, 18-FDG Positron Emission CT-scan (PET-CT), video-assisted thoracoscopy and resumed with the last TNM classification [17]. To rule out a suspected contralateral hilar or mediastinal nodal disease, bronchoscopy with endobronchial or oesophageal ultrasound (EBUS/EUS) was performed. In suspected diaphragmatic involvement, contrast-enhanced MRI was performed. 

Each patient was evaluated twice by our Multidisciplinary Tumour Board (MTB), i.e., at the histological diagnosis and then, after the iCT, before the curative-intent surgical procedure. The choice of the multimodal treatment was based on patients’ performance status (ECOG 0-1), comorbidities (adequate cardiorespiratory function, absence of renal insufficiency), clinical TNM stage (disease confined in a hemithorax, no bulky disease) and histological subtype (only epithelioid MPM). Patients deemed suitable for multimodality treatment were offered pre-operative chemotherapy and, in case of partial response or stable disease at re-staging, underwent P/D 4 to 5 weeks after the end of induction chemotherapy. Treatment protocols were approved by the institutional review board and informed consent was obtained from each patient before each treatment step. Our institutional review board granted approval and waived the requirement for specific informed consent for this retrospective study.

### 2.1. Pre-Operative Chemotherapy Protocol

A platinum-based chemotherapeutic regimen was used as iCT. The standard of care for MPM, i.e., the association between cisplatin 75 mg/m^2^ and pemetrexed 500 mg/m^2^ [2], was used as the first choice of treatment. Patients who were cisplatin un-fit for other clinical conditions (i.e., ECOG PS > 1, major cardiac comorbidity or creatinine clearance <60 mg/mL/min), were treated with the combination of carboplatin area under the plasma concentration-time curve (AUC) of 5 mg/mL/min plus pemetrexed 500 mg/m^2^ [18,19]. Four to six cycles of iCT were administered intravenously every 21 days. All patients received folic acid and vitamin B12 supplementation as indicated. Clinical and laboratory evaluation of patients was performed before each cycle of chemotherapy and when clinically requested, as a common clinical practice.

### 2.2. Surgical Protocol

All patients were placed in the appropriate lateral decubitus position and operated on with one-lung ventilation; they had invasive arterial monitoring, continuous pulse oximetry, central venous and urinary catheter in every case. Prophylactic antibiotics (cefazolin) were administered before skin incision. We achieved a maximal cytoreduction and a complete macroscopical resection (MCR) in all patients through a muscle-sparing posterolateral thoracotomy at the seventh intercostal space. The extrapleural plane was dissected from the apical region of the chest first, then dorsally and finally the dissection was carried out on the diaphragm and pericardium. After pleurectomy, the lung decortication was performed with particular attention to the dissection in the interlobar fissure. Extended resection, including pericardium and diaphragm, was undertaken only if full-thickness infiltrated and if considered absolutely necessary to achieve a complete macroscopic resection; pericardium and diaphragm were reconstructed with biological mesh or expanded polytetrafluoroethylene (ePTFE) mesh. In every case, hilar and mediastinal (including internal mammary) lymph node sampling was performed. 

After P/D, the lung was re-expanded to check for macroscopic air-leaks. We accepted even a substantial air leak whenever the lung showed a complete re-expansion. Two 30 French standard chest tubes were inserted, and the chest was closed meticulously in order to obtain an airtight closure and to prevent spillage of the chemotherapeutic agent during HITOC. Then, HITHOC was started in the lateral decubitus with the lung deflated by first filling the pleural space with 2000–4000 mL (depending on the chest size) of saline solution through the tubes using the anterior one as an inflow catheter and the posterior one as an outflow catheter. When the temperature reached 42 °C, the cisplatin at the standard dose of 150 mg/mq was inserted in the closed circuit of Performer LRT (RAND, Medtronic, Medolla (MO, Italy)). After one hour of perfusion, the solution was completely removed and the chest cavity washed with saline solution for another ten minutes, the lung inflated and chest tubes connected with standard collection chambers. All patients were extubated in the operating theatre or in ICU within the first post-operative day. After 24/48 h of ICU monitoring and in absence of early post-operative complications, patients were transferred to the ward where they continued hydration intravenously or by mouth, prevention of venous deep thrombosis with low molecular weight heparin and started respiratory physiokinetic therapy.

### 2.3. Adjuvant Therapies and Follow-Up

According to our multimodal treatment, adjuvant chemotherapy was considered after surgery and HITHOC only in patients with residual pathological disease (i.e., micro- or macroscopic positive surgical margins). As for the iCT, a platinum-based chemotherapy was the regimen preferred also in adjuvant setting. Adjuvant radiotherapy or other adjuvant therapies were not considered in our treatment protocol. Subsequent follow-up included clinical and radiological evaluation every 3–4 months. Patients who experienced relapse of the disease were offered local treatment, such as radiotherapy or systemic therapy (chemotherapy or immunotherapy if indicated).

### 2.4. Statistical Analysis

Statistical analysis was performed using SPSS 24.0 software (SPSS Inc., Chicago, IL, USA). Continuous variables are expressed as median and range, and categorical variables were resumed with counts and percentages. The Kaplan–Meier method was used to calculate overall, progression-free survival (PFS) and disease-free survival (DFS). Overall survival (OS) was calculated from the beginning of the multimodal treatment (i.e., from first cycle of iCT) to death or date of the last follow-up (30 June 2021); progression free survival (PFS) was calculated from the start of multimodal therapy to date of disease relapse; disease free survival (DFS) was calculated for those patients who received a maximum cytoreductive surgery from the date of operation to the date of first evidence of recurrence.

## 3. Results

During the study period, 65 patients affected by MPM were evaluated by our institutional MTB; of these, only 17 patients with c-stage I-IIIA were considered fit enough and then selected for the intentional-to-treat multimodality treatment including surgery. All 17 patients completed at least the planned three cycles of iCHT, but five of them were excluded from surgery because of progression of disease (*n* = 4) or judged unfit for surgery (*n* = 1) due to renal impairment and poor cardiac function. Twelve patients without progression of disease at the restaging with whole-body CT scan and PET-CT were finally selected for surgery (*n* = 9 mRECIST partial response, *n* = 3 stable disease) and received P/D and HITHOC. Table 1 shows demographic, functional and laboratoristic features of the entire cohort of patients who completed the multimodal treatment. Six (50%) patients were male, the median age at diagnosis was 67.5 (47–78 years) years and hypertension and mild renal failure was observed in two patients, whereas one patient received chemotherapy followed by radical surgery for bladder cancer and another woman had well-compensated multiple sclerosis. 

Table 2 shows clinical and pathological stage, type of surgery and structures resected, dosage of intraoperative cisplatin and post-operative results. In two patients with impaired renal function, the intraoperative dosage of cisplatin during HITHOC was reduce to 100 mg/m^2^. Extended resection included diaphragm (*n* = 4), pericardium (*n* = 3) and phrenic nerve (*n* = 3). Both diaphragm and pericardium were reconstructed with prosthesis in order to prevent spillage of the cisplatin into pericardial sac or peritoneum. Median ICU stay and median hospital stay were 2 days (1–7) and 10.5 days (6–20 days), respectively. Post-operative 30-day and 90-day mortality were 0, and complications developed in *n* = 7 (58.3%) patients. In particular, *n* = 3 (25%) patients developed chylothorax and were treated conservatively, *n* = 3 (25%) had a prolonged air leak and *n* = 1 (8.3%) was bleeding, requiring surgery. Blood transfusion was administered post-operatively in five (41.6%) patients. Two patients (16.6%) were discharged in a post-operative rehabilitation centre for respiratory recovery.

Final histopathological examination showed that all patients were affected by epithelioid MPM, 11 had N0 disease and one had N1 involvement (one lymph node metastasis in the aorto-pulmonary window). According to the pathological AJCC MPM TNM staging system, 8th edition [17], one patient showed complete pathological response (T×N0), three patients stage IA (pT1N0), seven patients stage IB (pT3N0) and one patient stage IIIA (pT3N1). One patient received adjuvant chemotherapy with carboplatin and pemetrexed for residual disease.

At the median follow-up of 21 months (range 12–37 months), four patients (33.3%) had died for progression of MPM; three patients (25%) were alive and free of disease and five (41.7%) remained alive with recurrence. In detail, seven (75.5%) and three (25%) patients developed local and both local and distant recurrence, respectively.

The median estimated OS from the beginning of the treatments was 31 months (CI95% 27.24–34.75) with 1-year, 2-year and 3-year OS of 100%, 100% and 55%, respectively (Figure 1A); the median OS from surgery was 26 months (CI95% 21.74–30.25) with 1-year, 2-year and 3-year OS of 100%, 56% and 37%, respectively (Figure 1B).

The median PFS was 24 months (CI95% 17.83–30.16) with 1-year and 2-year PFS of 92% and 28.5%, respectively (Figure 2A), whereas the median DFS was 19 months (CI95% 14.39–23.6) with 1-year and 2-year DFS of 83% and 0%, respectively (Figure 2B).

After the MPM relapse, five patients were treated with traditional chemotherapy, one entered in a study protocol with immunotherapy and two patients received radiation-therapy (IMRT: Intensity-Modulated Radiation Therapy and VMAT: Volumetric Modulated Arc Therapy).

The comparison between *n* = 5 patients initially excluded from surgery and *n* = 12 patients who underwent surgery showed a higher interval time between diagnosis and death for patients treated with the multimodal approach including surgery (15 months vs. 31 months, *p* = 0.26).

## 4. Discussion

Malignant pleural mesothelioma is an aggressive disease with a fatal outcome and poor survival for patients with the early-stage disease. MPM has some biological and anatomical characteristics that make a single treatment modality not adequate, and multimodality treatment strategies are currently being explored, with different modalities, timing and combinations without a definitive standard of care. Current guidelines from international societies support the role of surgery in multimodality concepts and approaches. This is because even in absence of strong evidence, real-life data from large population-based registries demonstrated a distinct survival advantage for patients undergoing multimodality treatments that include surgery [5,6]. The type of surgery with curative intent, always in a multimodal pathway, has been changed over time toward a more conservative approach, especially after the results of the MARS trial [9,20,21,22,23]. Nowadays, lung-sparing surgery is generally preferred to EPP, because it offers comparable long-term results with lower respiratory postoperative morbidity, lower risk of post-operative complications [9] and preservation of better quality of life [20]. Our experience demonstrates that P/D is safe even when applied in the setting of a complex multimodality approach including induction chemotherapy followed by surgery and HITHOC. We had no 30-day mortality and, although frequent, complications were not severe except for bleeding, which required a surgical revision. Furthermore, our multimodality treatment was well tolerated, as demonstrated by a low drop-out rate (*n* = 1/17, 6%) and the absence of any serious complication secondary to the pre-operative or intra-operative chemotherapy. The rationale behind neoadjuvant chemotherapy is to enhance the local and distant control, reducing the amount of tumour volume, in order to increase the macroscopic complete resection rate, which is the ultimate goal of surgery [3,11,21]. Moreover, this approach has some advantages: (1) pre-operative chemotherapy is better tolerated compared to adjuvant setting because the patient is not debilitated from surgery, (2) it is useful in the selection process, permitting to exclude patients who do not tolerate the treatment or develop disease progression, avoiding unnecessary surgery (we selected for surgery-only patients who showed a stable disease or a mRECIST response [24] and (3) lastly, we speculate that patients with good response to the pre-operative platinum-based chemotherapy could also have a good response to HITHOC due to a good susceptibility to the chemotherapeutic agent. Moreover, the increased risk of post-operative complication and mortality reported by several retrospective and prospective studies dealing with induction chemotherapy followed by EPP is not associated with lung-sparing surgery [20,23,25,26]. Despite this rationale, the use of preoperative chemotherapy is not very common, especially in North America [5,6,25], probably due to the potential of increased complications and mortality rate. Recently, Verma et al. [27], in their retrospective analysis of the National Cancer Data Base on 361 patients undergoing multimodality treatment of MPM, found a trend of increased mortality (*p* = 0.06) for patients undergoing induction chemotherapy followed by surgery compared to patients treated with upfront surgery followed by chemotherapy. The authors also described a decreasing use over time of the induction chemotherapy in the USA, probably for the concomitant reduction in performing EPP [21,25,26]. 

Even all the international guidelines, in absence of randomised controlled trials comparing upfront surgery (P/D) versus neoadjuvant approach, suggest to indifferently use either neoadjuvant or adjuvant chemotherapy [5,6]. To bring a contribution to the debate, a phase II study from EORTC (NCT02436733) currently randomises patients undergoing extended P/D between induction and adjuvant chemotherapy, and it is recruiting well in six centres [28]. As disease relapse and progression is local in most cases, there is a strong rationale in investigating and applying different forms of local therapy in order to improve the local control [12,13,14,15,16].

Thanks to the increasing and promising experience in the treatment of advanced peritoneal disease (ovarian carcinomatosis and other malignancies), the local treatment with hyperthermic chemotherapy has also been applied to MPM with the aim of maximising the effect of the cytoreductive surgery. The Hyperthermic Intraoperative THOracic Chemotherapy (HITHOC) acts on different ways to promote the killing of MPM cells: (1) the direct cytotoxicity of the platinum or other agents used, (2) the hyperthermia enables a deeper penetration of the chemotherapeutic agent into the tissues and (3) hyperthermia promotes cell apoptosis due to the increased susceptibility of neoplastic cells to the thermic injury [12,13]. Most published experiences showed low peri-operative, HITHOC-associated mortality [15,29,30,31] even if complications have been frequent, ranging from 65% [29] to 3.9% [15,31,32]. The most frequent HITHOC-related complications are a mild to moderate renal failure and deep venous thrombosis that can be effectively prevented with pre-operative fluid administration and active prophylactic administration of low molecular weight heparin [15]. Our experience shows that P/D and HITOC after induction therapy is a safe treatment strategy, as highlighted by zero 30-day mortality and not severe, although frequent, postoperative complications. 

Long-term outcomes of HITHOC in the multimodal treatment of MPM are sparse. Some encouraging results were reported in 2013 by Sugarbaker et al. [12], who retrospectively analysed a selected cohort of low-risk early-stage MPM patients who underwent macroscopic complete resection through EPP or P/D with or without the use of HITOC after surgery. They showed a significantly better median DFS (27.1 vs. 12.8 months, *p* = 0.0084) and OS (35.3 vs. 22.8 months, *p* = 0.028) in patients undergoing HITHOC. A phase I trial of combination cisplatin-gemcitabine HITHOC [32] on 104 MPM patients showed an OS of 20.3 months and a DFS of 10.7 months with 70% of recurrence located in the ipsilateral chest. Our multimodality treatment approach for early-stage MPM started in 2017, though long-term results are limited; with a median follow-up of 21 months, we observed an estimated median survival of 26 months and a disease-free interval of 19 months—outcomes in line with those published in previous studies and definitely better than those obtained with other treatments without surgery [23]. In fact, these results come from an accurate patients’ selection for surgery from the time of diagnosis, obtained with video-assisted thoracoscopy, up to the restaging phase (we used both CT-scan and PET-CT-scan) with a second MTB evaluation. 

The long-term results of lung-preserving surgery (extended P/D and P/D) are effectively reported in a systematic review by Cao et al. [7], showing a median OS for extended P/D ranging from 11.5 to 31.7 months and from 8.3 to 26 months for P/D, respectively. Data about DFS and recurrence varied across all the studies, with the median DFS ranging from 7.2 to 16 months for extended P/D and from 6 to 7.4 months for P/D. Comparing our data with these results, we can consider our survival outcomes significant, and, in particular, we achieved a substantial estimated DFS (19 months) that could be considered the results of the direct cytotoxic action of cisplatin during HITHOC.

Our study has several limitations, including the small sample size as a result of an aggressive therapeutic protocol applied to a highly-selected cohort of patients, which precludes a statistical analysis useful to identify any predictive factor of OS and DFS. Other limitations include the short follow-up period and the lack of a control group of patients with similar characteristics treated without surgery, but MPM is a rare neoplasm that makes it difficult to build randomised controlled trials and prospective studies, as demonstrated by the inconclusive results of the MARS trial and other RCTs on MPM.

## 5. Conclusions

In conclusion, our prospective study reported the safety and effectiveness of the multimodal treatment of early-stage epithelioid MPM, consisting of neoadjuvant platinum-based chemotherapy followed by P/D and HITHOC, because it was well-tolerated with a low treatment drop-out, no post-operative mortality and promising mid-term results.

## Figures and Tables

**Figure 1 jcm-10-05542-f001:**
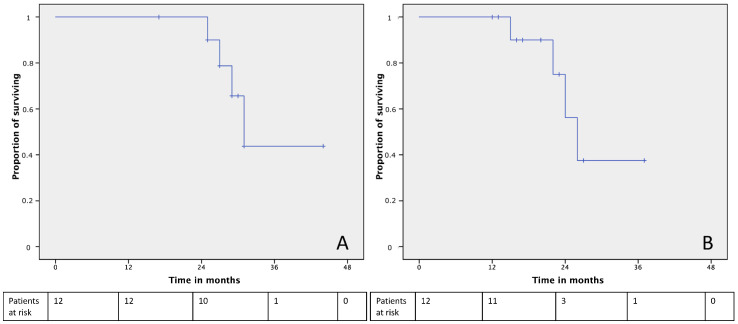
Overall survival curves from the histopathological diagnosis (**A**) and surgery (**B**).

**Figure 2 jcm-10-05542-f002:**
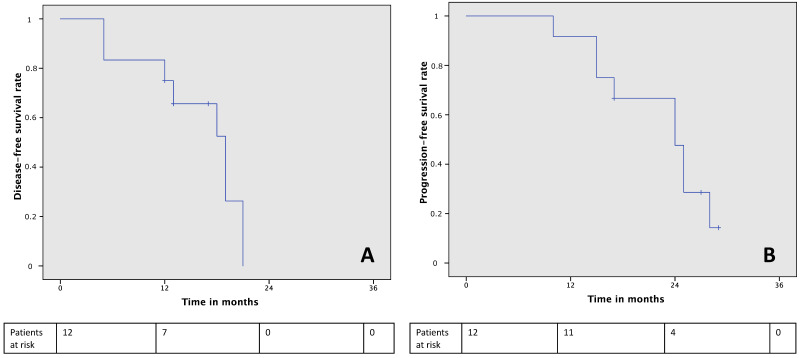
Disease-free survival curve (**A**) and progression-free survival curve (**B**).

**Table 1 jcm-10-05542-t001:** Demographic, functional and laboratoristic data.

Variable	Median or Number (%)	Range
Age	67.5	47–78
Sex male	6 (50%)	
BMI	25.5	22–31
Asbesto expusure	9 (75%)	
PS ECOG		
0	9 (75%)	
1	3 (25%)	
CCI index	4	2–7
MPM subtype		
Epithelioid	12 (100%)	
Pre-op FEV1%	89	68–111
Pre-op FVC%	90.5	75–126
Pre-op DLCO%	70.5	48–99
Pre-op RBC (×10^12^/L)	4.01	3.52–5.4
Post-op RBC (×10^12^/L)	3.42	2.76–5.2
Pre-op Hb (g/dL)	12.35	9.9–14.6
Post-op Hb (g/dL)	10.9	8.8–13.6
Pre-op WBC (×10^9^/L)	5.83	4.27–8.53
PLT pre (×10^9^/L)	238	114–346
Pre-op Creatinine (mg/dL)	0.82	0.62–1.13
Post-op Creatinine (mg/dL)	0.78	0.56–1.3
Pre-op eGFR	93.5	75–134
Post-op eGFR	91	76–120

BMI: Body Mass Index; PS ECOG: Performance Status by Eastern Cooperative Oncology Group; CCI: Charlson Comorbidity Index; MPM: Malignant Pleural Mesothelioma; FEV1%: Forced Expiratory Volume at 1 second; FVC%: Forced Vital Capacity; DLCO% Diffusing Capacity for Carbon Monoxide; RBC: Red Blood Cells; WBC: White Blood Cells; PLT: Platelets; eGFR: Estimated Glomerular Filtration Rate.

**Table 2 jcm-10-05542-t002:** Clinical and pathological stage, type of surgery and structures resected, dosage of intraoperative cisplatin and post-operative results.

Variable	*n* (%)	Median (Range)
Side		
Right	8 (66.6%)	
Left	4 (33.3%)	
Structures clinically involved		
P	4 (33.3%)	
P, D	3 (25%)	
P, V	2 (16.6%)	
P, V, D	1 (8.4%)	
P, V, D, Pe	2 (16.6%)	
Staging TNM		
cT1N0M0	5 (41.6%)	
cT2N0M0	6 (50%)	
cT3N0M0	1 (8.4%)	
Clinical stage		
IA	5 (41.6%)	
IB	7 (58.4%)	
Type of NAC		
CDDP + pemetrexed	5 (41.6%)	
CBDCA + pemetrexed	7 (58.4%)	
Type of surgery/structures resected		
P/D	6 (50%)	
eP/D diaphragm	3 (25%)	
eP/D pericardium	2 (16.6%)	
eP/D diaphragm+pericardium	1 (8.4%)	
Cisplatin dosage	150 mg/mq	100–150 mg/mq
Pathological involvement		
CR	1 (8.4%)	
Major pathological response, P	1 (8.4%)	
Major pathological response, V	1 (8.4%)	
P, V	1 (8.4%)	
P, V, D	2 (16.6%)	
P, V, D, Pe	2 (16.6%)	
P, V, lung, ln5	1 (8.4%)	
P, V, D, Pe, lung	2 (16.6%)	
P, V, D, lung, pericardial fat	1 (8.4%)	
pTNM		
CR	1 (8.4%)	
pT1N0M0	3 (25%)	
pT3N0M0	7 (58.4%)	
pT3N1M0	1 (8.4%)	
Pathological stage		
CR	1 (8.4%)	
IA	3 (25%)	
IB	7 (58.4%)	
IIIA	1 (8.4%)	
ICU stay	2 days	1–7 days
Hospital stay	10.5 days	6–20 days
Complications in detail		
Blood transfusion	5 (58.4%)	
Prolonged air leak	3 (25%)	
Chylothorax	3 (25%)	
Bleeding requiring surgery	1 (8.4%)	
Post-operative re-habilitation	2 (16.6%)	
Chronic pain	1 (8.4%)	
Recurrence		
No recurrence	3 (25%)	
Local	6 (50%)	
Local and distant	3 (25%)	

P: Parietal; V: Visceral; D: Diaphragm; Pe: Pericardium; TNM: Tumor, Node and Metastasis; NAC: Neoadjuvant therapy; CBDCA: carboplatinum; CDDP: cisplatinum; P/D: pleurectomy and decortication; eP/D: extended pleurectomy and decortication; ln5: lymph node station 5; CR: Complete pathological Response; ICU: Intensive Care Unit.

## Data Availability

The data presented in this study are available on request from the corresponding author.

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
