# Peer review of "Induction Chemotherapy Followed by Pleurectomy Decortication and Hyperthermic Intraoperative Chemotherapy (HITHOC) for Early-Stage Epitheliod Malignant Pleural Mesothelioma—A Prospective Report"

_jcm, 2021, doi:10.3390/jcm10235542_

Round 1

Reviewer 1 Report

I read very carefully your paper and I found it interesting.

Appears a well developed and executed work.

Nevertheless, I would like for some metholodological and technical specifications.

  • For the unfit cisplatin patients during neoadjuvant treatment; your choose equally cisplatin for the HITHOC?  Can you specify the rationale of your choice.
  • Maintaining deflated the lung during perfusion it's an good choice; what do you thing about partial ventilation, when low air leak present after decortication. Can reduce the global  liquid volume perfusion and increase concentration of chemotherapic drugs.
  • As regards the methods I would like to have a comment of yours. I fully agree with the three points  between lines 258-268, regarding neoadjuvant chemo, though can delay surgery and/or increase peri-operative complications. Considering that relapse site was mainly local, in your cohort; have you evaluated a re-do HITHOC during the same hospitalization stay? do you have any data of adjuvant systemic chemotherpy as a completion of multimodal treatment? do you plan any further studies for a future comparison?.

Author Response

REVISION OF THE PAPER: jcm-1464733 considered by Journal of Clinical Medicine

Title: INDUCTION CHEMOTHERAPY FOLLOWED BY PLEURECTOMY DECORTICATION AND HYPERTHERMIC INTRAOPERATIVE CHEMOTHERAPY (HITHOC) FOR EARLY-STAGE EPITHELIOD MALIGNANT PLEURAL MESOTHELIOMA, A PROSPECTIVE REPORT

Dear Editor,

We would like to thank the reviewers for their thoughtful review of the manuscript. They raise important issues and their inputs are very helpful for improving the manuscript. We agree with almost all their comments and we have revised our manuscript accordingly. We hope that the reviewers will find our responses to their comments satisfactory.

We are confident that the new version of the manuscript will be greatly improved.

As you suggested, we reply to each comment in point-by-point fashion.

The modified text has been inserted with Word track changes.

Thank you very much for your effort.

Reviewer 1:

I read very carefully your paper and I found it interesting.

Appears a well developed and executed work.

Nevertheless, I would like for some metholodological and technical specifications.

  • For the unfit cisplatin patients during neoadjuvant treatment; your choose equally cisplatin for the HITHOC?  Can you specify the rationale of your choice.
  • Maintaining deflated the lung during perfusion it's an good choice; what do you thing about partial ventilation, when low air leak present after decortication. Can reduce the global  liquid volume perfusion and increase concentration of chemotherapic drugs.
  • As regards the methods I would like to have a comment of yours. I fully agree with the three points  between lines 258-268, regarding neoadjuvant chemo, though can delay surgery and/or increase peri-operative complications. Considering that relapse site was mainly local, in your cohort; have you evaluated a re-do HITHOC during the same hospitalization stay? do you have any data of adjuvant systemic chemotherpy as a completion of multimodal treatment? do you plan any further studies for a future comparison?.

Comment 1: For the unfit cisplatin patients during neoadjuvant treatment; your choose equally cisplatin for the HITHOC?  Can you specify the rationale of your choice.

Answer 1: Thank you for your comment. In our cohort n=2 had an impaired renal function and then we decided to reduce the intraoperative cisplatin dose up to 100 mg/mq. These two patients went well with this low dosage and they did not develop a post-operative renal failure. Our decision was based on previous study that showed that the absorption of the chemotherapeutic agents (cisplatin) is low and the plasmatic levels after HITHOC are very low as showed by some papers (Ried M, Potzger T, Braune N, Diez C, Neu R, Sziklavari Z, Schalke B, Hofmann HS. Local and systemic exposure of cisplatin during hyperthermic intrathoracic chemotherapy perfusion after pleurectomy and decortication for treatment of pleural malignancies. J Surg Oncol. 2013 Jun;107(7):735-40. doi: 10.1002/jso.23321. Epub 2013 Feb 5. PMID: 23386426.; van Ruth S, van Tellingen O, Korse CM, Verwaal VJ, Zoetmulder FA. Pharmacokinetics of doxorubicin and cisplatin used in intraoperative hyperthermic intrathoracic chemotherapy after cytoreductive surgery for malignant pleural mesothelioma and pleural thymoma. Anticancer Drugs. 2003 Jan;14(1):57-65. doi: 10.1097/00001813-200301000-00008. PMID: 12544259.).

Changes 1: No changes.

Comment 2: Maintaining deflated the lung during perfusion it's an good choice; what do you thing about partial ventilation, when low air leak present after decortication. Can reduce the global  liquid volume perfusion and increase concentration of chemotherapic drugs.

Answer 2: Thank you for your comment. The partial ventilation on the operated lung reduces the liquid input and then can reduce the possible complications of the liquid volume such as hemodinamical instability. On the other hand, the partial ventilation could decrease the distribution of the chemotherapeutic solution in every pleural recesses. To overcome the high volume of fluid that can lead to a hemodynamical instability, we slowly infuse the pleural cavity. As a consequence of the complete decortication, at the end of the procedure a substantial air leak is present that makes it difficult to obtain an adequate perfusion of the pleural space.

Changes 2: No changes

Comment 3: As regards the methods I would like to have a comment of yours. I fully agree with the three points  between lines 258-268, regarding neoadjuvant chemo, though can delay surgery and/or increase peri-operative complications. Considering that relapse site was mainly local, in your cohort; have you evaluated a re-do HITHOC during the same hospitalization stay? do you have any data of adjuvant systemic chemotherpy as a completion of multimodal treatment? do you plan any further studies for a future comparison?

Answer 3: Thank you for you interesting questions. We have not performed any re-HITHOC yet, but we will think about it in the future. We did not consider HITHOC for a relapse because after pleurectomy and decortication we do not have any space to infuse the solution. Regarding the adjuvant treatment, only one patient had adjuvant chemotherapy with carboplatinum and pemetrexed. Moreover, different treatments were used for patients who developed recurrence including radiation therapy, chemotherapy and immunotherapy. Actually, we do not have any other active study, but in the future the neoadjuvant chemo-immunotherapy regimen for the multimodal treatment of MPM could become a new topic of interest. 

Changes 3: No changes.

Reviewer 2 Report

Dear Editor and Authors,

Following the results of the Mesothelioma and Radical Surgery (MARS) trial which has currently lead to the abandonment of extrapleural pneumonectomy as a treatment procedure for mesothelioma, thoracic surgeons have been extensively investigating alternative, efficacious and less debilitating therapeutic surgery techniques!

Therefore it was with quite an interest that I read this study titled “Induction Chemotherapy followed by Pleurectomy Decortication and Hyperthermic Intraoperative Chemotherapy (HITHOC) for early stage Epitheloid Malignant Pleural Mesothelioma” by Dr. Bongiolatti and colleagues from the Careggi University Hospital in Florence, Italy.

In this retrospective analysis (and this needs to be clarified and corrected in the text, for example the first line of the conclusions) of database data in 12 patients with stage I or II epithelioid mesothelioma which underwent a multimodality therapy with chemotherapy, surgery and HITHOC the authors claim they have demonstrated favorable and efficacious long term results! I do have my concerns regarding this (please see comment number 3 below).

The analysis suffers from all the major limitations of such studies (as the authors have also alluded) that is a small patient number, selection bias, single institution origin of data and retrospective nature. Therefore, the amount of clinically useful conclusions that can be reached by it is limited and need to be considered with caution!

The manuscript is well written in clear and understandable language. On a section by section review:

The introduction is well written, concise and presents the current status well. I don’t think it needs any change.

The materials and methods section also describes well the study’s setup. I do have a couple of comments/queries:

  1. The period of the study in months should be listed not just “since 2017”
  2. A statement that the study was conducted according to established ethical criteria and that approval was obtained (even though individual patient consent may had been waved) from their institution’s ethics committee/board needs to be inserted in the beginning of the M &M section. The statement in line 99-100 has to do with treatment consent and not the study’s!
  3. Did all patients undergo PET CT for staging purposes?
  4. If PET CT was performed why was EBUS/EUS performed/necessary? Was it routinely performed for staging? Was it also accompanied by lymph node biopsies? If so, which stations were biopsied routinely? The whole sentence about EBUS/EUS reads incomplete, there is no verb and no action described! It needs rewriting/editing.
  5. If I understand it correctly not only parietal decortication but also visceral decortication was performed. Was this a complete decortication of the whole lung or only in sections with disease/thickening? The amount of air leak produced must have been significant (hence the two 30 Fr chest tubes – by the way I was not aware there were 30 Fr size drains available as I have been using 24, 28, 32 and 36 in my practice. Is this accurate?)
  6. How was HITHOC prevented from entering the systemic circulation/ getting absorbed if the decortication was performed first? I am worried about increased levels in the blood and adverse reactions? What is the authors view on the matter?

The results section is adequate but needs some modifications to make it more easily understood and present/transmit the information better to the reader. Specifically:

  1. Table 1 needs to be condensed. As it is now with individual patient data listed it is too big and cluttered and no information/evaluation of the overall group can be given. Please create a standard demographics table with means and SD or median values!
  2. Same for Table 2.
  3. The lack of control group(s) is a methodological problem as there is no possible comparison with standard treatment and therefore no useful assessment if this whole treatment process which is long, costly and burdensome/risky for the patient is actually worth it!! Why did the authors not include/compare their group with other patients who did not undergo this multimodality approach? Can the authors comment what the OS and PFS rates are in Italy and if their outcomes are better than the reported national averages?
  4. Line 255 is just a question! Was it a comment by one of the authors during the final review stage of the manuscript which was left there unintentionally?

The discussion section is well written and presents and discusses well the available literature and known information. I don’t really have much comments about it.

In conclusion therefore, this study has major limitations!! Even the authors agree to this!! However, it does offer some useful albeit almost anecdotal data about the use of this combined multimodality treatment approach and especially of the use of HITHOC for the treatment of epithelioid mesothelioma. As such I am inclined to consent to its presentation/publication on condition that they answer the above queries adequately!!

Thank you again for asking me to review this work. I wish all, and especially the authors well.

Kind regards,

Author Response

Reviewer 2: Dear Editor and Authors,

Following the results of the Mesothelioma and Radical Surgery (MARS) trial which has currently lead to the abandonment of extrapleural pneumonectomy as a treatment procedure for mesothelioma, thoracic surgeons have been extensively investigating alternative, efficacious and less debilitating therapeutic surgery techniques!

Therefore it was with quite an interest that I read this study titled “Induction Chemotherapy followed by Pleurectomy Decortication and Hyperthermic Intraoperative Chemotherapy (HITHOC) for early stage Epitheloid Malignant Pleural Mesothelioma” by Dr. Bongiolatti and colleagues from the Careggi University Hospital in Florence, Italy.

In this retrospective analysis (and this needs to be clarified and corrected in the text, for example the first line of the conclusions) of database data in 12 patients with stage I or II epithelioid mesothelioma which underwent a multimodality therapy with chemotherapy, surgery and HITHOC the authors claim they have demonstrated favorable and efficacious long term results! I do have my concerns regarding this (please see comment number 3 below).

The analysis suffers from all the major limitations of such studies (as the authors have also alluded) that is a small patient number, selection bias, single institution origin of data and retrospective nature. Therefore, the amount of clinically useful conclusions that can be reached by it is limited and need to be considered with caution!

The manuscript is well written in clear and understandable language. On a section by section review:

The introduction is well written, concise and presents the current status well. I don’t think it needs any change.

The materials and methods section also describes well the study’s setup. I do have a couple of comments/queries:

  1. The period of the study in months should be listed not just “since 2017”
  2. A statement that the study was conducted according to established ethical criteria and that approval was obtained (even though individual patient consent may had been waved) from their institution’s ethics committee/board needs to be inserted in the beginning of the M &M section. The statement in line 99-100 has to do with treatment consent and not the study’s!
  3. Did all patients undergo PET CT for staging purposes?
  4. If PET CT was performed why was EBUS/EUS performed/necessary? Was it routinely performed for staging? Was it also accompanied by lymph node biopsies? If so, which stations were biopsied routinely? The whole sentence about EBUS/EUS reads incomplete, there is no verb and no action described! It needs rewriting/editing.
  5. If I understand it correctly not only parietal decortication but also visceral decortication was performed. Was this a complete decortication of the whole lung or only in sections with disease/thickening? The amount of air leak produced must have been significant (hence the two 30 Fr chest tubes – by the way I was not aware there were 30 Fr size drains available as I have been using 24, 28, 32 and 36 in my practice. Is this accurate?)
  6. How was HITHOC prevented from entering the systemic circulation/ getting absorbed if the decortication was performed first? I am worried about increased levels in the blood and adverse reactions? What is the authors view on the matter?

The results section is adequate but needs some modifications to make it more easily understood and present/transmit the information better to the reader. Specifically:

  1. Table 1 needs to be condensed. As it is now with individual patient data listed it is too big and cluttered and no information/evaluation of the overall group can be given. Please create a standard demographics table with means and SD or median values!
  2. Same for Table 2.
  3. The lack of control group(s) is a methodological problem as there is no possible comparison with standard treatment and therefore no useful assessment if this whole treatment process which is long, costly and burdensome/risky for the patient is actually worth it!! Why did the authors not include/compare their group with other patients who did not undergo this multimodality approach? Can the authors comment what the OS and PFS rates are in Italy and if their outcomes are better than the reported national averages?
  4. Line 255 is just a question! Was it a comment by one of the authors during the final review stage of the manuscript which was left there unintentionally?

The discussion section is well written and presents and discusses well the available literature and known information. I don’t really have much comments about it.

In conclusion therefore, this study has major limitations!! Even the authors agree to this!! However, it does offer some useful albeit almost anecdotal data about the use of this combined multimodality treatment approach and especially of the use of HITHOC for the treatment of epithelioid mesothelioma. As such I am inclined to consent to its presentation/publication on condition that they answer the above queries adequately!!

Thank you again for asking me to review this work. I wish all, and especially the authors well.

Kind regards,

Thank you for your valuable comments and suggestions. Our study population is small due to an extensive selection and I’m completely agree with you regarding the results that can be interpreted with caution. I modified the paper following your comments.

Section Materials and Methods

Comment 1: The period of the study in months should be listed not just “since 2017”.

Answer 1: Thank you for your suggestion. I changed the sentence as you suggested.

Changes 1: P3L80: From January 2017 to December 2020…

Comment 2: A statement that the study was conducted according to established ethical criteria and that approval was obtained (even though individual patient consent may had been waved) from their institution’s ethics committee/board needs to be inserted in the beginning of the M &M section. The statement in line 99-100 has to do with treatment consent and not the study’s!

Answer 2: I’m agree with you. I inserted a sentence regarding the institutional review board.

Changes 2: P4L104-106: Our institutional review board granted approval and waived the requirement for specific informed consent for this retrospective study.

Comment 3: PET CT for staging purposes?

Answer 3: Every patients had the PET-CT scan pre-treatment for staging purpose and if a nodal hyperactivity was highlighted, the EBUS/EUS was performed to assess the nodal status.

Changes 3: No changes.

Comment 4: If PET CT was performed why was EBUS/EUS performed/necessary? Was it routinely performed for staging? Was it also accompanied by lymph node biopsies? If so, which stations were biopsied routinely? The whole sentence about EBUS/EUS reads incomplete, there is no verb and no action described! It needs rewriting/editing.

Answer 4: EBUS/EUS was selectively performed based on the results of the PET-CT-scan whenever a nodal activity was highlighted and we need biopsies to rule out a nodal disease.

Changes 4: P3 L89-90: To rule out a suspected contralateral hilar or mediastinal nodal disease, bronchoscopy with endobronchial or oesophageal ultrasound (EBUS/EUS) was performed.

Comment 5: If I understand it correctly not only parietal decortication but also visceral decortication was performed. Was this a complete decortication of the whole lung or only in sections with disease/thickening? The amount of air leak produced must have been significant (hence the two 30 Fr chest tubes – by the way I was not aware there were 30 Fr size drains available as I have been using 24, 28, 32 and 36 in my practice. Is this accurate?)

Answer 5: Thank you for your comment. We routinely performed a complete removal of the visceral (on whole lung) and parietal pleura and in few cases we did some pulmonary wedge resection due to an extensive involvement of the lung. Due to the visceral pleura removal, we observed a high incidence of air leak at the end of the surgical procedure. However, only two patients developed prolonged air leak and these two patients were sent home with the chest tube in place attached to an Heimlich valve. We think that two large bore tubes are adequate for P/D being effective both for air and blood. The 30F chest tube is completely available in our institute.

Changes 5: No changes.

Comment 6: How was HITHOC prevented from entering the systemic circulation/ getting absorbed if the decortication was performed first? I am worried about increased levels in the blood and adverse reactions? What is the authors view on the matter?

Answer 6: We did not use any medication or “trick” to prevent the systemic absorption of the cisplatin. As showed by the previous literature the cisplatin dose that we used is considered because of low systemic absortion and low plasmatic levels (Ried M, Potzger T, Braune N, Diez C, Neu R, Sziklavari Z, Schalke B, Hofmann HS. Local and systemic exposure of cisplatin during hyperthermic intrathoracic chemotherapy perfusion after pleurectomy and decortication for treatment of pleural malignancies. J Surg Oncol. 2013 Jun;107(7):735-40. doi: 10.1002/jso.23321. Epub 2013 Feb 5. PMID: 23386426.; van Ruth S, van Tellingen O, Korse CM, Verwaal VJ, Zoetmulder FA. Pharmacokinetics of doxorubicin and cisplatin used in intraoperative hyperthermic intrathoracic chemotherapy after cytoreductive surgery for malignant pleural mesothelioma and pleural thymoma. Anticancer Drugs. 2003 Jan;14(1):57-65. doi: 10.1097/00001813-200301000-00008. PMID: 12544259.). Furthermore, we did not observe any acute kidney injury even in patient with previous renal failure (n=2).

Changes 6: No changes.

Results

Comment 1: Table 1 needs to be condensed. As it is now with individual patient data listed it is too big and cluttered and no information/evaluation of the overall group can be given. Please create a standard demographics table with means and SD or median values!

Answer 1: Thank you for your comment, I’m agree with you and so I changed the demographic table as you suggested, resuming data with median and range (due to the small number of cases and not normally distributed), frequencies and percentages.

Changes 1: I modified and appropriately inserted this table:

Variable

Median or number (%)

Range

Age

67.5

47-78

Sex male

6 (50%)

BMI

25.5

22-31

Asbesto expusure

9 (75%)

PS ECOG

0

1

9 (75%)

3 (25%)

CCI index

4

2-7

MPM subtype

Epithelioid

12 (100%)

Pre-op FEV1%

89

68-111

Pre-op FVC%

90.5

75-126

Pre-op DLCO%

70.5

48-99

Pre-op RBC  (x10^12/L)

4.01

3.52-5.4

Post-op RBC (x10^12/L)

3.42

2.76-5.2

Pre-op Hb   (g/dL)

12.35

9.9-14.6

Post-op Hb (g/dL)

10.9

8.8-13.6

Pre-op WBC (x10^9/L)

5.83

4.27-8.53

PLT pre (x10^9/L)

238

114-346

Pre-op Creatinine   (mg/dL)

0.82

0.62-1.13

Post-op Creatinine   (mg/dL)

0.78

0.56-1.3

Pre-op eGFR

93.5

75-134

Post-op eGFR

91

76-120

Comment 2: Same for Table 2.

Answer 2: Thank you for your comment, I’m agree with you and so I changed the results table as you suggested.

Changes 2: I modified and appropriately insert this table.

Variable

n (%)

Median (range)

Side

Right

Left

8 (66.6%)

4 (33.3%)

Structures clinically involved

P

P, D

P, V

P, V, D

P, V, D, Pe

4 (33.3%)

3 (25%)

2 (16.6%)

1 (8.4%)

2 (16.6%)

Staging TNM

cT1N0M0

cT2N0M0

cT3N0M0

5 (41.6%)

6 (50%)

1 (8.4%)

Clinical stage

IA

IB

5 (41.6%)

7 (58.4%)

Type of iCT

CDDP+pem

CBDCA+pem

5 (41.6%)

7 (58.4%)

Type of surgery/structures resected

P/D

eP/D diaphragm

eP/D pericardium

eP/D diaphragm+pericardium

6 (50%)

3 (25%)

2 (16.6%)

1 (8.4%)

Cisplatin dosage

150 mg/mq

100-150 mg/mq

Pathological involvement

CR

Major pathological response, P

Major pathological response, V

P, V

P, V, D

P, V, D, Pe

P, V, lung, ln5

P, V, D, Pe, lung

P, V, D, lung, pericardial fat

1 (8.4%)

1 (8.4%)

1 (8.4%)

1 (8.4%)

2 (16.6%)

2 (16.6%)

1 (8.4%)

2 (16.6%)

1 (8.4%)

pTNM

CR

pT1N0M0

pT3N0M0

pT3N1M0

1 (8.4%)

3 (25%)

7 (58.4%)

1 (8.4%)

Pathological stage

CR

IA

IB

IIIA

1 (8.4%)

3 (25%)

7 (58.4%)

1 (8.4%)

ICU stay

2 days

1-7 days

Hospital stay

10.5 days

6-20 days

Complications in detail

Blood transfusion

Prolonged air leak

Chylothorax

Bleeding requiring surgery

Post-operative re-habilitation

Chronic pain

5 (58.4%)

3 (25%)

3 (25%)

1 (8.4%)

2 (16.6%)

1 (8.4%)

Recurrence

No recurrence

Local

Local and distant

3 (25%)

6 (50%)

3 (25%)

Comment 3: The lack of control group(s) is a methodological problem as there is no possible comparison with standard treatment and therefore no useful assessment if this whole treatment process which is long, costly and burdensome/risky for the patient is actually worth it!! Why did the authors not include/compare their group with other patients who did not undergo this multimodality approach? Can the authors comment what the OS and PFS rates are in Italy and if their outcomes are better than the reported national averages?

Answer 3: During the study period, we selected all potentially eligible patients (n=17) to be treated for elarly stage MPM with our protocol including HITHOC. Then, all the other patients excluded from thi protocol are not comparable with the patients included in this study in terms of pre-operative characteristics. Regarding the n=17 patients selected for this treatment protocol, n=5 were excluded because of disease progression during neoadjuvant chemotherapy or because the patients were considered unfit for surgery. We analyzed and compared the survival from diagnosis to death between these n=5 patients with n=12 patients treated with the multimodal approach including surgery and we had a high survival for the “surgical arm” (15 months vs 31 months, log rank test p=0.26). This datum was not showed before because we think that the comparison was full of biases, but in this version we can included it because it could demonstrate an impact of our multimodal approach on survival.

Changes 3: we add this sentence P9L254-257: “The comparison between n=5 patients initially excluded from surgery with n=12 patients who underwent surgery showed a higher interval time between diagnosis and death for patients treated with the multi-modal approach including surgery (15 months vs 31 months, p=0.26).”

Comment 4: Line 255 is just a question! Was it a comment by one of the authors during the final review stage of the manuscript which was left there unintentionally?

Answer 4: Yes, you are right it was an authors’ comment during the final whole review about a possible “control group”.

Changes 4: I deleted the sentence in P10L244-245.
